# Optimization of a Six-Step Pressure Swing Adsorption Process for Biogas Separation on a Commercial Scale

**Bundit Kottititum [1,\*], Thongchai Srinophakun [2], Niwat Phongsai [3] and Quoc Tri Phung [4,\*]**

[1] Center of Excellence on Petrochemical and Materials Technology, Chulalongkorn University, Bangkok 10330, Thailand

[2] Department of Chemical Engineering, Kasetsart University, Bangkok 10900, Thailand; fengtcs@ku.ac.th

[3] RE Power Service Co., Ltd., Bangkok 10900, Thailand; niwat@repower.co.th

[4] Institute for Environment, Health, and Safety, Belgian Nuclear Research Centre (SCK CEN), 2400 Mol, Belgium

\* Correspondence: bundit.k@ku.th (B.K.); quoc.tri.phung@sckcen.be (Q.T.P.); Tel.: +66-849-855-100 (B.K.); +32-1433-3240 (Q.T.P.)

**Abstract:** Pressure swing adsorption (PSA) appears to be an effective technology for biogas upgrading under different operating conditions with low greenhouse gas emissions. This study presents the simulation of biomethane adsorption with the adsorption bed filled with a carbon molecular sieve (CMS). A six dual-bed six-step PSA process was studied which produced a high purity of biomethane. The design of the adsorption bed was followed by the real process of which the biomethane capacity was more than 5000 Nm$^3$/h. For the adsorbent, a CMS-3K was used, and a biomethane gas with a minimum 92% purity was produced at 6.5 bar adsorption pressure. To understand the adsorption characteristics of the $CH_4$ and $CO_2$ gases, the Langmuir isotherm model was used to determine the isotherm of a mixed gas containing 55% $CH_4$ and 45% $CO_2$. Furthermore, the experimental data from the work of Cavenati et al. were used to investigate the kinetic parameter and mass transfer coefficient. The mass transfer coefficients of two species were determined to be 0.0008 s$^{-1}$ and 0.018 s$^{-1}$ at 306 K for $CH_4$ and $CO_2$, respectively. The PSA process was then simulated with a cyclic steady state until the relative tolerance was 0.0005, which was then used to predict the $CH_4$ and $CO_2$ mole fraction along the adsorption bed length at a steady state. Moreover, the optimal conditions were analyzed using Aspen Adsorption to simulate various key operating parameters, such as flowrate, adsorption pressure and adsorption time. The results show a good agreement between the simulated results and the real operating data obtained from the company REBiofuel. Finally, the sensitivity analysis for the major parameters was presented. The optimal conditions were found to be an adsorption pressure of 6 bar, an adsorption time of 250 s and a purity of up to 97.92% with a flowrate reducing to 2000 Nm$^3$/h. This study can serve as a commercial approach to reduce operating costs.

**Keywords:** pressure swing adsorption; simulation; carbon molecular sieve; biogas; biomethane

---

## 1. Introduction

Nowadays, the process of extracting cassava starch requires a large amount of water. This water ultimately becomes wastewater, with an amount of approximately 12 m$^3$ per ton of starch. There will be approximately 1.4 tons of solid waste. Most of the wastewater is caused by the washing and peeling processes in the rotating tank. This wastewater has a low oxygen demand (COD) [1]. However, another wastewater is produced from draining the sedimentation starch tank, which has a high COD and

biochemical oxygen demand (BOD) [2]. Therefore, treating this wastewater is an important expense of cassava flour factories.

Normally, the wastewater from cassava starch is commonly used to produce biogas by using the anaerobic digestion method, which produces biogas containing ~55% $CH_4$ and ~45% $CO_2$. Biogas with more than 50% $CH_4$ can directly produce electricity. Moreover, the biogas can also be used for transportation, but it is necessary to have a methane concentration of at least 92%, also known as biomethane [3]. The upgrading processes for $CO_2$ removal include water scrubbing, membrane separation and pressure swing absorption (PSA) [4]. It is worth mentioning that Sircar et al. designed the first PSA unit (GEMINI) for commercial scale upgrading of biogas. They were successful in recovering landfill gas to high purity methane, which led to a US patent in 1988 [5]. A comparison between the various upgrading technologies is summarized in Table 1. It appears that the PSA technique has a low operating cost and methane loss, which is a key factor to reduce the greenhouse gas [6]. The PSA is a well-known separation technique which uses the adsorbent and changing pressure to separate the selected gases. The amount of adsorption gas depends on the operating pressure and the equilibrium capacity of the adsorbent, including the adsorption kinetics. The first two-bed PSA process was proposed by Skarstrom [7], which consists of a four-step process and is known as the Skarstrom cycle. The core cycle consists of a pressurization step, an adsorption step, a depressurization or a blowdown step and a purge step.

**Table 1.** The inventory data of biomethane upgrading processes adapted from [1].

| Cleaning and Upgrading | Key Parameter | Average Data |
|---|---|---|
| Water scrubbing (WS) | Electricity<br>Upgrading yield (88% Methane)<br>Methane losses | 0.20 kWh/N m$^3$ biogas<br>68%<br>5.13% |
| Pressure swing absorption (PSA) | Electricity<br>Upgrading yield (91% Methane)<br>Methane losses | 0.24 kWh/N m$^3$ biogas<br>65%<br>4.00% |
| Membrane separation (MS) | Electricity<br>Upgrading yield (91% Methane)<br>Methane losses | 0.19 kWh/N m$^3$ biogas<br>65%<br>6.50% |

Many researchers have focused their studies on the steps of the PSA process and process conditions. Cavenati et al. [8], for example, compared the counter current and co-current of a four-step process which has both adiabatic and non-adiabatic conditions. The results showed that the counter and co-current were almost the same in both conditions, but the process performance was slightly better for the adiabatic than the non-adiabatic condition [8]. Park et al. [9] later found a method which performed the best energy saving and productivity by comparing a four-step process proposed by Skarstrom [7] with five-step and six-step processes that included the rinse step and pressure equalization step. The results showed that the purity of the six-step process could be increased without a significant increase in the specific power consumption.

There are a variety of widely used adsorbents to adsorb carbon dioxide in biogas, such as zeolite, a carbon molecular sieve (CMS), and activated carbon [4]. Grande and Rodrigues [10] studied the performance of the CMS-3K adsorbent in a five-step process with the zeolite adsorbent in a four-step process. The results showed that both the adsorbents obtained a purity of $CH_4$ higher than 98%, but the recovery of $CH_4$ of CMS-3K was higher than zeolite, i.e., 80% compared to 60% of zeolite. Due to the microporous adsorbents, the pores on a micro- and macro-level of the CMS were distinguishable and dispersed in the pore structure. The pores at a molecular size in the CMS have provided a high kinetic selectivity and adsorption capacity for various gases [11,12]. Therefore, the CMS is chosen in this study because it allows for a higher level of carbon dioxide diffusion in the micro-pore network than methane molecules, which have a strong resistance on the surface and inside the micro-pores of the adsorbent.

Moreover, the previous studies simulated the PSA process using experimental data at a lab scale for a simulated gas. As a consequence, it is difficult to scale-up for commercial processes. In this study, we focused on the commercial scale PSA unit at the company REBiofuel in Thailand, which has the six-step separation process and CMS adsorbent. The PSA unit was simulated by using an extended Aspen Plus software known as Aspen Adsorption®. The objective of the current study is to validate of the model with experimental breakthrough data. The simulated results are then compared with operating data to optimize the working conditions.

## 2. Materials and Methods

The data for dynamic simulation in this research were obtained by the Aspen Adsorption® simulation program (version V11, Aspen Technology, Inc., Bedford, MA, USA). Besides that, the mathematical program MATLAB® (version R2018a, MathWorks, Natick, MA, USA) was used to fit some groups of experimental data in this study in order to optimize the adsorbent properties. In addition, the experimental data obtained from the study of Cavenati et al. [8] and REBiofuel were used to feed into the simulation program interface as references in order to validate the simulation model and apply to real processes.

### 2.1. Pressure Swing Adsorption Steps

The PSA process has many steps, of which the pressure changing inside the adsorption bed, the step time and the valve acting sequence for each step are different. Normally, four operation steps are performed in one cycle of the PSA process. In this study, the pressure equalization steps (including depressurization equalization step (DPE) and pressure equalization step (PPE)) were added to a basic four-step process proposed by Skarstrom to improve process efficiency, as shown in Figure 1.

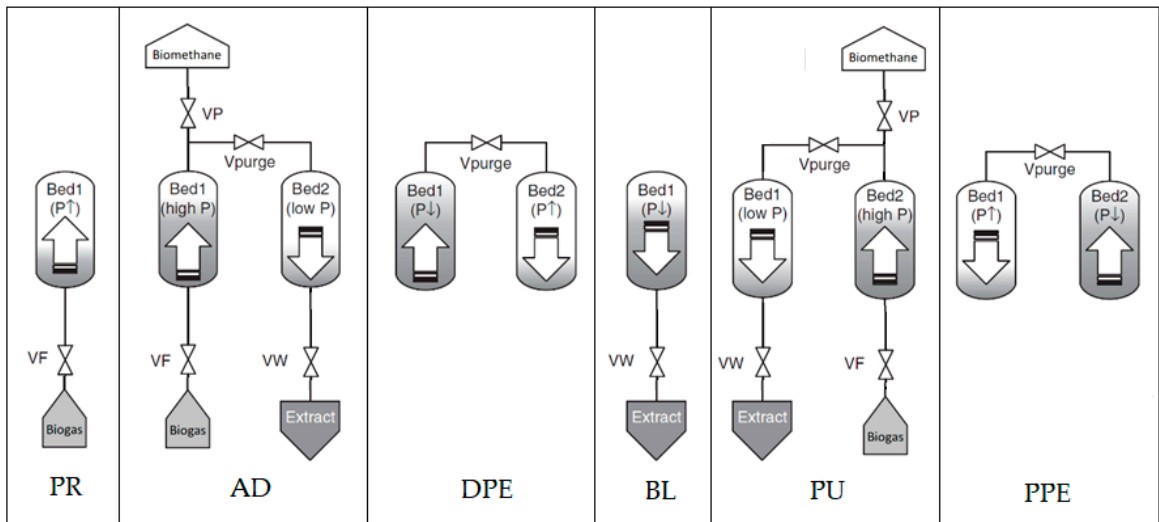

**Figure 1.** The improved Skarstrom cycle in the pressure swing adsorption (PSA) process—symbols are defined in the text.

First, biogas flows into the adsorption bed (Bed1) and increases the pressure to the desired pressure (PR). The process enters the adsorption step (AD) where the second bed contains more $CO_2$ and the outlet stream of Bed1 is enriched in less $CO_2$ components (high $CH_4$). Then, a portion of $CH_4$ product is moved to Bed2 to assist in the regeneration of the adsorbent. After that, Bed1 is loaded and Bed2 is regenerated. The depressurizing equalization step (DPE) is then taken place. This step is important to reduce energy consumption due to the quantity of the gas to be compressed. After the equalization step, the process repeats with the reversion of the two bed roles. The basic process includes the following six steps [13]:

1. Pressurization step, PR;
2. Adsorption step, AD;
3. Depressurization equalization step, DPE;
4. Blowdown step, BL;
5. Purging step, PU;
6. Pressure equalization step, PPE.

According to the real process, twelve adsorption columns are used, in which six coupled adsorption and desorption pairs work in parallel. Therefore, the inlet biogas flowrate is equal to 1/6 of each adsorption column.

*2.2. Thermodynamic Model*

As a biogas flows through the adsorption bed, the volume is changed due to pressure gradients, and the gas velocity as well as the temperature are also affected. Moreover, these parameters also influence the pressure drop along the adsorption bed. Therefore, we need to carefully select the thermodynamic model in order to correctly generate the volumetric flowrate of the gas phase. For those reasons, in this study, the Peng–Robinson model was selected, which is suitable for the non-ideal gas phases and is widely used for biogas separation with the PSA technique [14].

*2.3. Flowsheet*

The interacting single bed full flowsheet can be used to simulate full cyclic systems of interacting units. The flowsheet shows the use of interacting in the single bed approach. The approach is only valid for the following assumptions:

- Each adsorbent bed is identical (adsorbent layers, model assumptions);
- Only one bed has to be rigorously modeled;
- Any number of interactions can be incorporated;
- Material sent to an interacting bed is reused (replayed) later in the cycle.

The interaction unit needs to store the information of the material sent to the interacting bed, such as flowrate, composition, temperature and pressure. The final flowsheet is shown in Figure 2, where product valve (VP), feed valve (VF) and purge valve (VW) are ramp valves; BED is the reference bed; D1 is interacting bed representation; VD is a fictitious valve necessary to connect the simulated bed rigorously to the D bed, and it is able to replicate the opposite of what occurs in BED. The biogas stream is the main feed which has 55% $CH_4$ and 45% $CO_2$ in this study. The biomethane stream is the main product.

*2.4. The Properties of Packed Beds*

2.4.1. Material and Momentum Balance

In order to make the simulation model to be as close as possible to reality, particular attention was paid to the configuration of the bed of the adsorbent material. It was considered a uniform empty fraction in the entire column. Furthermore, we have chosen to simplify the treatment by assuming a piston flow (convection only and zero dispersion, therefore a Peclet number tending to infinity) only developed along the axial coordinate. The momentum balance adopted is based on the Ergun equation, expressed as [15]:

$$\frac{\partial P}{\partial z} = \frac{1.5 \cdot 10^{-3} \mu}{d_p^2} \frac{(1-\varepsilon)^2}{\varepsilon^2} U + \frac{1.75 \cdot 10^{-5} \rho_g}{d_p} \frac{(1-\varepsilon)}{\varepsilon} U^2 \tag{1}$$

where $P$ is pressure (bar), $z$ is the fixed bed height (m), $\mu$ is the dynamic fluid viscosity, $\rho_g$ is the gas density (kg/m$^3$), $d_p$ is particle diameter (m), $U$ is gas velocity (m/s), and $\varepsilon$ (-) is the porosity.

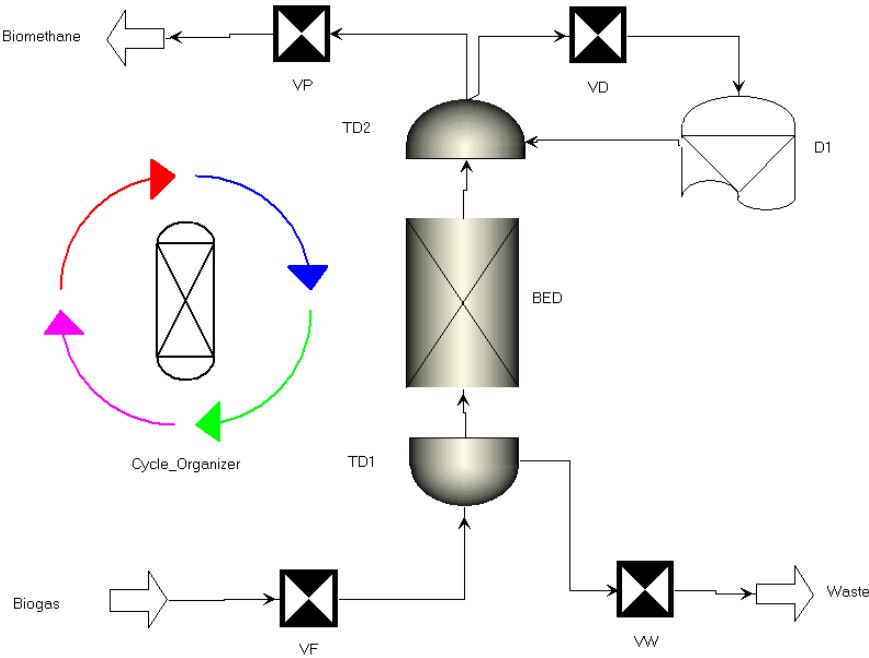

**Figure 2.** Scheme of the PSA process simulated in Aspen Adsorption.

### 2.4.2. Adsorption Isotherm

As for the adsorption isotherm, the Langmuir 3 model was used, expressed as a function of the partial pressure of the various components presented in Aspen Adsorption and able to take into account the dependence on temperature. It can be expressed as follows [16]:

$$q_i = \frac{(\text{IP}_{1,i} + \text{IP}_{2,i}\text{T})\left(\text{IP}_{3,i} \exp \frac{\text{IP}_{4,i}}{\text{T}}\right)p_i}{1 + \left(\text{IP}_{3,i} \exp \frac{\text{IP}_{4,i}}{\text{T}}\right)p_i} \tag{2}$$

where $q_i$ is the concentration of component $i$ in the adsorbent material (kmol/kg), T is temperature (K), and $p_i$ is partial pressure of component $i$ (bar). The $\text{IP}_i$ are the regressed parameters, whose values are regressed with MATLAB®. It should be noted that these parameters might not have a physical meaning like the ones in the original Langmuir 3 model.

### 2.4.3. Specify the Kinetic Model Assumption and Mass-Transfer Coefficients

In addition to the determination of the parameters of the adsorption isotherm, the correct description of the kinetics is of fundamental importance for an adsorbent material such as the CMS. In this case, it was decided to adopt a lumped resistance model (concentrated resistance) with a linear shape, which has the capability of simplifying the treatment as much as possible when taking into account a single resistance per component. Various authors [17] affirm that the diffusion of $CH_4$ and $CO_2$ in carbon molecular sieves can be separated into three different contributions: diffusion within the macropores, resistance to the entrance of the microporous and diffusion inside the micropores. However, Cavenati et al. [8] reported that the contribution of the resistance to the entrance of the micro-pore is negligible in the case of $CO_2$ at the typical working pressures of the PSA process. Furthermore, on the basis of what is reported in the Aspen Adsorption Guide, it is clear that for the CMS materials, the limiting resistance is constituted by the diffusion contribution in the micropores. For this reason, as already mentioned, it was decided to simplify the treatment by taking a single

kinetic parameter for $CH_4$ and $CO_2$. The kinetics is fitted with the linear driving force models that are typically used in kinetic studies [18]. The linear driving force model can be written as follows [19]:

$$\frac{\partial q_i}{\partial t} = K_i\left(q_i^* - q_i\right) \tag{3}$$

where, $i$ = 1, 2 for $CO_2$ and $CH_4$, respectively, $K_i$ is the mass transfer coefficient ($s^{-1}$), $q_i$ is the adsorbed amount of component $i$ (kmol/kg), and $q_i^*$ is the equilibrium concentration of the adsorbed phase (kmol/kg).

### 2.4.4. Energy Balance

In this work, the energy balance was set as isothermal. Note that in reality, the temperature may vary up to 20 K as shown in the work of Choi [20], which is also the case of this study. However, we used the average temperature and considered isothermal conditions, as this allows us to completely ignore the energy balance. The gas temperature and the solid temperature are held constant and equal.

### 2.4.5. Carbon Molecular Sieve (CMS)

For the adsorbent, REBiofuel used a carbon molecular sieve which has properties similar to that of the CMS-3K of Takeda Corporation [8]. The main data are summarized in Table 2.

**Table 2.** Physical properties of CMS-3K.

| Parameter Name | Value | Unit |
|---|---|---|
| Adsorbent bulk solid density | 715 | kg/m$^3$ |
| Adsorbent particle radius | $9 \times 10^{-4}$ | m |
| Porosity | 0.33 | - |

### *2.5. The Cycle Organizer*

To create cyclically operated processes, a tool, namely Cycle Organizer (Figure 2), has been provided. The purpose of the Cycle Organizer is to allow the user to define an unlimited number of cycle descriptions and enable the user to determine an unlimited number of steps in a cycle by defining the cyclic task. In this study, we need to fix the step time to that of REbiofuel conditions in order to verify the model. Then, we optimized the maximum flowrate of biogas that meets the standard purity of biomethane.

## 3. Results and Discussion

### *3.1. Determination of the Model Parameters*

#### 3.1.1. Langmuir Isotherm

For the simulation model, the most important parameters are isotherm and kinetic data of the biogas component, which are assumed to include only $CH_4$ and $CO_2$ in this study. The separation of mixed gas occurs due to the differences in selectivity of the adsorbent at the same pressure. Moreover, the difference in adsorption equilibrium and kinetics also affects the selectivity of mixed gas [21]. We used the Langmuir isotherm model to determine the adsorption equilibrium, expressed via Equation (2). The least square method in MATLAB$^{®}$ was used to determine the isotherm parameters ($IP_i$) of $CH_4$ and $CO_2$ by minimizing the curve between experimental data from Cavenati et al. [8] and the Langmuir isotherm equation.

The function of the regression, which makes the solution closely match the experimental data, is expressed as follows [22]:

$$MIN \sum_i \left[ F(x, x_{model,i}) - y_{exp,i} \right]^2 \text{ where } x = [IP_1, \dots, IP_4] \tag{4}$$

The adsorption equilibrium curves of $CH_4$ and $CO_2$ obtained by the Langmuir isotherm equation of $CH_4$ and $CO_2$ and fitted with experimental data are shown in Figure 3.

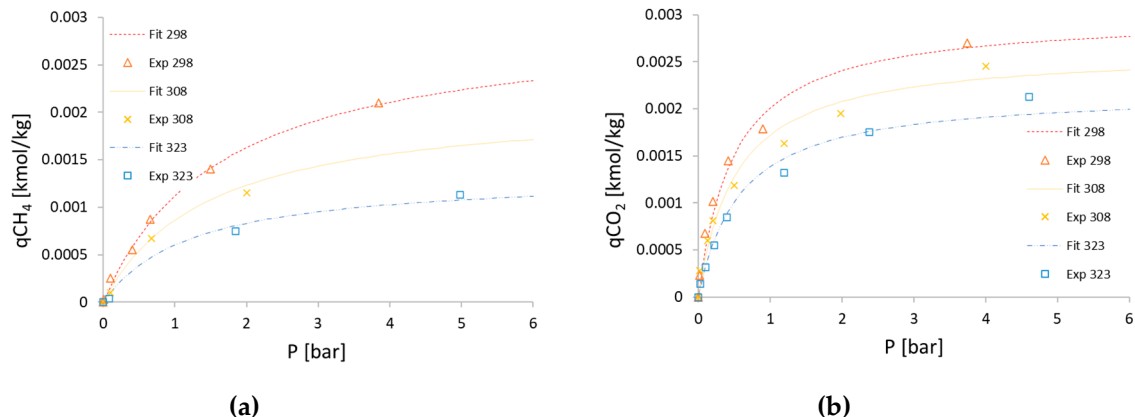

**(a)**           **(b)**

**Figure 3.** Regression of the $CH_4$ (**a**) and $CO_2$ (**b**) adsorption isotherm from experimental data at various temperatures with MATLAB (Langmuir 3 isotherm).

The regressed curves lead to a slight underestimation of the $CO_2$ adsorption balance at pressures above 2.5 bar. The Langmuir isotherm parameters obtained for $CH_4$ and $CO_2$, corresponding to Type I (among the five types) as identified by Brunauer [23], are summarized in Table 3. It should be noted that these values, to which the experimental data are available, have no physical meaning outside the temperature range (298–323 K).

**Table 3.** Langmuir isotherm parameters.

| Descriptions | IP$_1$ ($\times 10^{-6}$) | IP$_2$ | IP$_3$ | IP$_4$ |
|:---:|:---:|:---:|:---:|:---:|
| $CH_4$ | 2.876 | 1918 | 28.77 | −1152 |
| $CO_2$ | 12.74 | 1839 | 0.2486 | 626.8 |

### 3.1.2. Mass-Transfer Coefficients

The rigorous transport model is needed to numerically simulate the PSA process for the separation of biogas containing $CH_4$ and $CO_2$. The mass transfer coefficient and the equilibrium adsorption model are the major parameters for verifying the modelling results with experimental data. These parameters were regressed starting from the experimental concentration as a function of time at 308 K, as provided by Cavenati et al. [8]. In this case, $c_0 = P_0/RT_0$ ($P_0$ = 1 atm, $T_0$ = 308 K). The linear driving force model can be derived into terms of concentration via the following equation:

$$c_i = c_0 - \varepsilon \cdot q_i \tag{5}$$

It is possible to derive $q_i$ as

$$q_i^* - q_i = \frac{c_0 - c_i(t)}{\varepsilon} \tag{6}$$

By replacing Equation (6) in Equation (3)

$$\frac{\partial c_i(t)}{\partial t} = K_i(c_0 - c_i(t))$$

(7)

It is possible to derive the equation to solve the $K_i$ of methane and carbon dioxide from the available data. The result of the regression is shown in Figure 4.

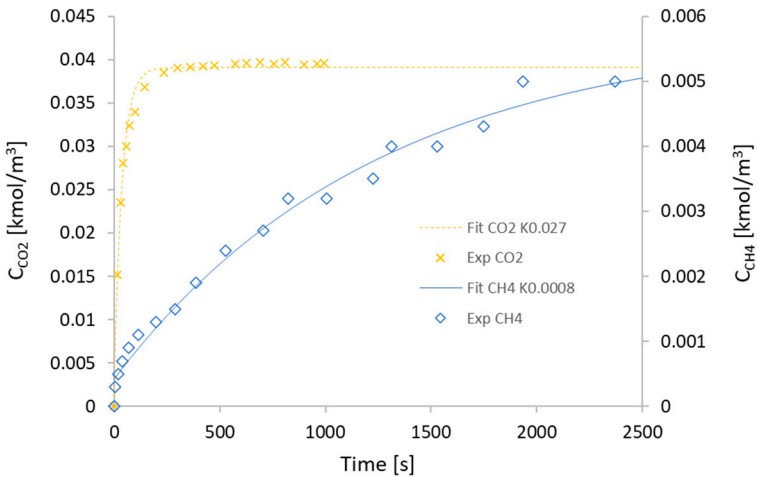

**Figure 4.** $CH_4$ and $CO_2$ concentrations as a function of time.

According to the parametric analysis results, it is found that the modelling results are fitted very well with the experimental data, and the mass transfer coefficient of $CO_2$ is 0.027 s$^{-1}$, which is larger than that for $CH_4$ of 0.0008 s$^{-1}$. It can be seen that there is a significant difference between these two kinetic parameters. The reason is that when the biogas is injected in the adsorption bed, most of the $CH_4$ is released through the outlet without adsorption. This is due to the low selectivity of the adsorbent, which is related to a very low mass transfer coefficient. However, in the case of $CO_2$, the large amount of $CO_2$ was adsorbed due to fast adsorption kinetics.

*3.2. Validation of the Model*

To verify the correctness of the assumptions explained in Section 3.1, it is useful to simulate a simple breakthrough curve of a $CO_2/CH_4$ mixture with 45% vol./55% vol. proportions, in order to reproduce the data obtained from an experiment by Cavenati et al. involving a fixed bed track [8]. An inlet pressure of 3.2 bar and a temperature of 303 K have been set, with a total flow rate equal to 1 standard liter per minute (SLPM). The same dimensions with Cavenati's study were maintained for the adsorbent bed, i.e., a height of 0.83 m and a diameter of 0.021 m.

In order to function, Aspen Adsorption must initialize the gas content inside the adsorbent bed. Since it was not possible to set the column to be empty at the initial conditions, it was decided to assume that the bed was completely filled with $N_2$, requiring that it could not be adsorbed (isotherm parameters = 0) and did not offer any resistance to diffusion ($K_{N2} = 0$). In this way, a curve quite similar to the experimental data was obtained in Figure 5, but with an excessive slope regarding the $CO_2$ breakthrough curve. This is related to the fact that, while the kinetic parameters of methane and carbon dioxide have regressed to 308 K, the experimental temperatures were obtained at 303 K. This is because of the fact that the slope of the curve is not only changed due to the adsorption limit of adsorbent but also the temperature increase during the adsorption step, resulting in the non-symmetric S-shape curve due to a faster diffusion. However, the isothermal operation is presented in this study, so the thermal effects are lumped into the mass transfer coefficient, which is helpful in obtaining a faster solution. For this reason, manually decreasing the kinetic parameter of $CO_2$ allowed us to obtain $K_{CO2}$

= 0.01 s$^{-1}$, which was a better fitting of the curve. The results of the two simulations, first keeping the $K_{CO2}$ regressed to 308 K and then correcting it to 303 K, are shown in Figure 5 for (K0.01) and (K0.027).

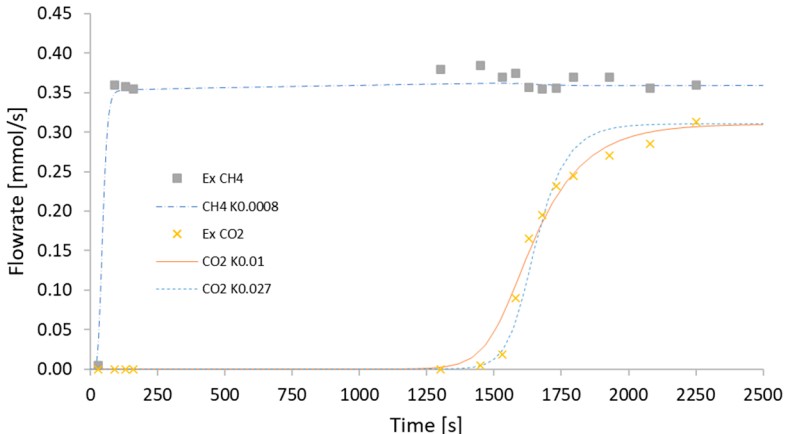

**Figure 5.** Breakthrough curves obtained with Aspen Adsorption compared with the experimental data of Cavenati et al. [8], (- - -) curve of $CH_4$ obtained with $K_{CH4}$ = 0.0008 s$^{-1}$ (·····) curve of $CH_4$ obtained with $K_{CO2}$ = 0.027 s$^{-1}$, (——) curve obtained with $K_{CO2}$ = 0.01 s$^{-1}$.

As can be seen from the graphs, the starting point of the experimental data of $CO_2$ is linked to the adsorption isotherm and is therefore not changed. Until 1250 s, the slope of the curve is changed due to the adsorption limit of the adsorbent and is related to the influence of the mass transfer coefficient. Thus, once the value of the kinetic parameter of $CO_2$ was corrected at a temperature of 303 K, we proceeded with a fitting with MATLAB$^{®}$ to derive the temperature coefficient of $K_{CO2}$ in an exponential form of the Arrhenius equation:

$$K_{CO2} = k_0 \cdot e^{\left(-\frac{E}{RT}\right)} \tag{8}$$

where R, $k_0$, $E$ are the ideal gas constant, pre-exponential factor, and the activation energy, respectively, and T is temperature. In this way, we obtained $k_0$ = 1.89·10$^{27}$ s$^{-1}$, $E$ = 1.70 ×·10$^5$ J/mol. This dependence of the kinetic coefficient of $CO_2$ on temperature has been included in Aspen Adsorption and used for subsequent simulations.

### 3.3. Comparison Model with Real Operating Data

Once the validity of the model was verified, it was applied for real PSA process simulations. The base case was developed with a supply of 5000 Nm$^3$/h of biogas of agricultural origin (45% vol. $CO_2$, 55% vol. $CH_4$) that reproduced the specifications made by the manufacturer as much as possible. In this case, the purity of the biomethane was acceptable and equals 92%. The reason for this decrease in purity is linked to the feed flowrate, adsorption time and adsorption pressure of the real process. All of these parameters influence the purity and recovery of $CH_4$. Moreover, the decrease in $CH_4$ recovery is particularly important considering its high greenhouse gas emission.

It is useful at this point to clarify the definitions of purity and recovery of methane adopted in the following equations [10]:

$$Purity = \frac{\int_0^{t_{ads}} c_{CH4} v_g \big|_{z=H} dt}{\int_0^{t_{ads}} c_{CH4} v_g \big|_{z=H} dt + \int_0^{t_{ads}} c_{CO2} v_g \big|_{z=H} dt} \tag{9}$$

$$Recovery = \frac{\int_0^{t_{ads}} c_{CH4} v_g \big|_{z=H} dt - \int_0^{t_{spurgo}} c_{CH4} v_g \big|_{z=H} dt}{\int_0^{t_{ads}} c_{CH4} v_g \big|_{z=H} dt} \tag{10}$$

In this case, $t_{ads}$ and $t_{spurgo}$ represent, respectively, the times of the adsorption and purging phases, $c_{CH4}$ and $c_{CO2}$ are the concentrations in (mol/m$^3$) of methane and $CO_2$, respectively, $v_g$ is the speed of the gas phase moving in the column (m/s) and $H$ (m) represents the height of the bed. These are, in fact, integral means of the parameters over time since the flow rates and purities in the various currents are not constant. From the data provided by REBiofuel, it has been observed that, in general, six columns are used to treat 5000 Nm$^3$/h of biogas. It was therefore decided to divide the total supply flow rate by six, assuming that the columns work in pairs. The results presented below are therefore intended for a single pair of columns, which is fed a flow rate of 37.17 kmol/h, corresponding to 1/6 of the total.

The chosen adsorption pressure of 6.5 bar followed the suggestion from REBiofuel, while the one set for desorption is equal to 0.5 absolute bar due to the steep slope of the carbon dioxide adsorption isotherm, which would not allow a substantial release at higher operating pressures. A temperature of 306 K was set, intermediate between 301 and 311 K, among which the behavior of the mass transfer coefficient of $CO_2$ was regressed, and it equaled 0.018 s$^{-1}$. The dimensions of the column were initially hypothesized to correspond to a scale-up with a ratio of fed biogas flow rate/volume of the column constant with respect to the conditions of the study by Cavenati et al. [8]. The height to diameter ratio was set to five, in accordance with the work of Grande and Rodrigues [10]. In those studies, they used the condition of purity of at least 98% and, based on this, the estimated bed design from Aspen Adsorption should be have a diameter and height of 0.698 and 3.49 m, respectively, resulting in a very long adsorption time of around 700 s. In the real process, it is required only 92% purity, and REBiofuel indicated that the maximum duration of the cycle is ten minutes, thereby indicating a probable oversizing of the column. For this reason, there was a slight decrease in the volume used for adsorption, while the same height/diameter ratio was maintained. A height of 3.00 m and a diameter of 0.60 m were set for the simulated process, and these values are actually the same as those in the real process, as show in Table 4. The velocity of gas can be calculated from the data of the surface (A, in m$^2$) and volumetric input flow rate. It is possible to obtain the initial speed of the gas phase along the column, as show in the following equation:

$$v_{g,in} = \frac{V}{A \cdot \varepsilon_i} \tag{11}$$

where $A \cdot \varepsilon_i$ represents the free section of the column (m$^2$) and $V$ is the volumetric flowrate (m$^3$/s). In this way, a gas flow velocity of 0.18 m/s was obtained. Note that values of feed surface speed lower than or close to 0.2 m/s are able to allow a reasonable solid–gas contact and to reduce the effects of axial dispersion [12].

**Table 4.** Bed design used in simulation.

| Parameter Name | Value | Unit |
|----------------|-------|------|
| Bed length | 3.00 | m |
| Bed inner diameter | 0.60 | m |
| Bed outer diameter | 0.608 | m |
| Bed porosity | 0.33 | - |

We then need to determine the step time of the pressurization step (PR), the adsorption step (AD), the depressurizing equalization step (DPE), the purging step (PU), the blowdown step (BL) and the pressure equalization step (PPE). For the pressurization step, we can calculate it from the flowrate to increase the pressure until 6.5 bar and, in this case, the pressurization step time is 25 s. Then, in order to decide the time of the adsorption phase, 15 cycles at different (long) adsorption times were processed in advance to observe that the biological methane concentration could be reduced at any point in time. The result is shown in Figure 6.

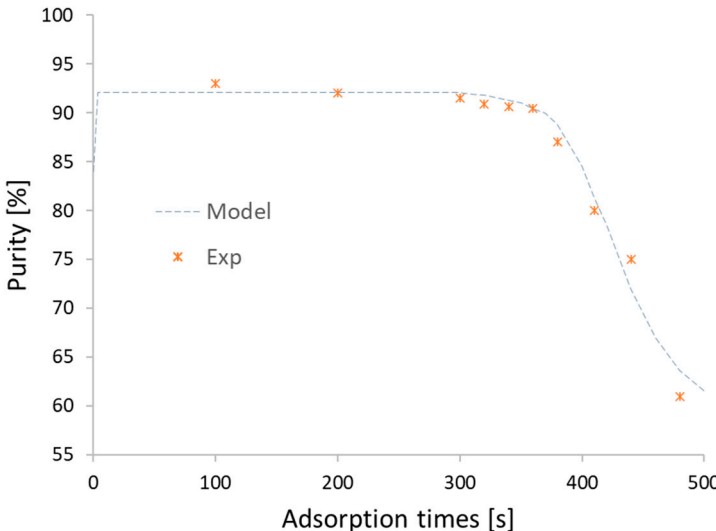

**Figure 6.** Effect of time on the purity of the $CH_4$ output. Result after 15 cycles and by varying the time of the adsorption phase to verify the correctness of the result.

It can be observed that the curves obtained at the initial adsorption period have a 92% purity until 300 s, after which the molar fraction of $CH_4$ begins to drop. The effect of the adsorption time on purity and recovery has therefore been investigated with greater precision by varying the duration of the adsorption phase (and consequently, of the regeneration phase) shorter than 300 s. The program was iterated until the cyclic steady state (CSS) was reached, at which time the process performance was stable over time. For this purpose, a relative tolerance has been set for the closure of the material and the energy balances equal 0.0005. It has been noted that CSS is achieved after 94 iteration cycles, as shown in Table 5. This is due to the slowness of the $CH_4$ adsorption process. In fact, it takes a long time for the amount of methane adsorbed inside the CMS to stabilize.

**Table 5.** Biomethane purity, recovery and number of cycles necessary to reach the cyclic steady state (CSS) as a function of the adsorption time.

| Case | Simulation | Real Conditions |
|---|---|---|
| Pressure (bar) | 6.5 | 6.5 |
| PR (s) | 25 | 25 |
| AD (s) | 250 | 250 |
| DPE (s) | 5 | 5 |
| BL (s) | 25 | 25 |
| PU (s) | 250 | 250 |
| PPE (s) | 5 | 5 |
| Recovery of $CH_4$ (%) | 81.63 | - |
| Purity of $CH_4$ (%) | 92.08 | 92.10 |
| Number of cycles | 94 | - |

Having obtained a good agreement of the simulation and real process, it was decided to adopt an adsorption time of 250 s for a total cycle duration of 560 s (9.3 min). This allows us to obtain a biomethane with a purity slightly higher than 92% molar. Moreover, in this way, the cycle time is similar to the operating conditions of REBiofuel. The simulation of the pressure cycle was compared with the data from the real process, as shown in Figure 7.

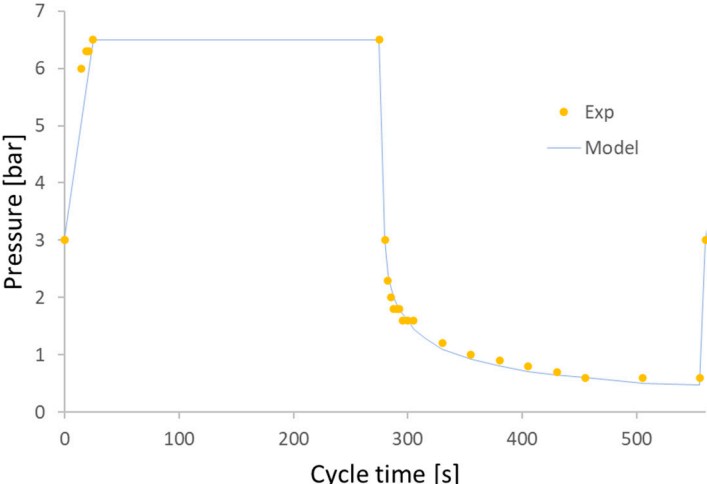

**Figure 7.** The pressure in the adsorbent bed as a function of time, dotted symbols represent the real operating condition, while the continuous curve shows the simulated results from Aspen Absorption.

The molar flowrates of the product and purge valve, respectively, are shown in Figure 8. Regarding the adsorption and desorption steps, when the product valve is opened, there is an almost instantaneous exit of a peak flow, mainly consisting of methane accumulated in the column during the pressurization phase. Then, the output flow remains almost constant until the end of the adsorption phase (PS $CH_4$ and $CO_2$). Following the opening of the purge valve during the blowdown, there is a further peak in the flow rate of $CH_4$ (W $CH_4$). As a result of the adsorption phase, there is a free part of $CH_4$ in the bed because $CO_2$ is still in the adsorbent. However, there is also a peak in the $CO_2$ flow rate at the outlet (W $CO_2$). Although the molar flowrate of $CO_2$ is lower than $CH_4$ at an earlier time, it begins to gradually come out with a lower flowrate in the desorption phase. A zero flowrate is not reached at the end, indicating that the regeneration of the material is not complete. Although it is difficult to notice from the graph, in the purge phase, the flow of $CH_4$ is certainly not zero. In fact, the small portion of biomethane produced is used to facilitate desorption in the vacuum.

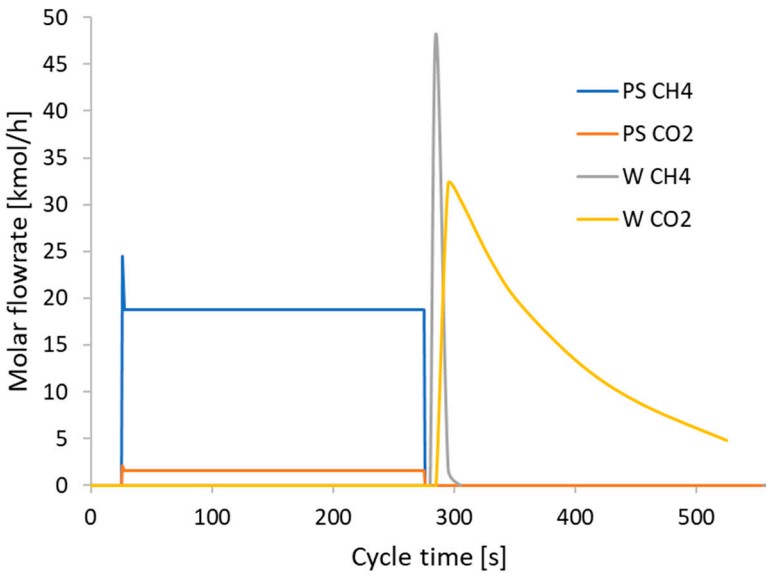

**Figure 8.** Trend of molar flow rates of $CO_2$ and $CH_4$ during the cycle time in the product stream (PS $CH_4$ and $CO_2$) and waste stream (W $CH_4$ and $CO_2$) at the last cycle.

The $CH_4$ and $CO_2$ loads adsorbed in the CMS at the end of each step are presented in Figures 9 and 10. These results refer to cycle 94, in which a CSS situation was reached. Figure 9 shows that the amount of $CH_4$ adsorbed along the adsorption bed remains substantially constant through the cycle and slightly high on the top of adsorption bed, as $CO_2$ is first absorbed. Therefore, the proportion of $CH_4$ rises on the top of the adsorption bed and the adsorption amount of $CH_4$ is also increased proportionally. On the other hand, the $CO_2$ load varies significantly during the various phases, as shown in Figure 10. The adsorbed amount of $CO_2$ at the end of the adsorption step is high, but at the blowdown and purge steps, $CO_2$ comes out from the adsorbent due to the regeneration step. This is precisely linked to the kinetic nature of the process.

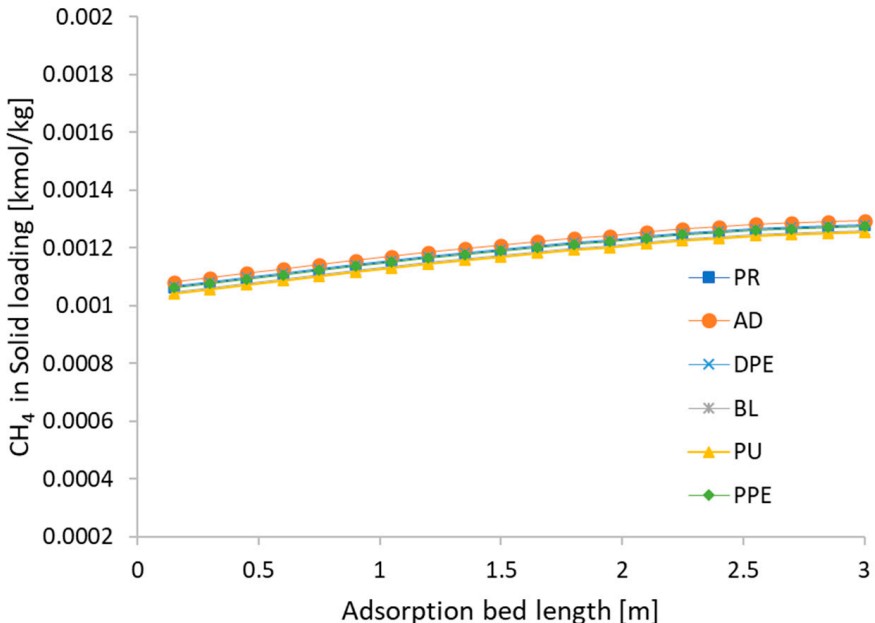

**Figure 9.** Quantity of $CH_4$ adsorbed along the column at the end of the various phases (cycle 94).

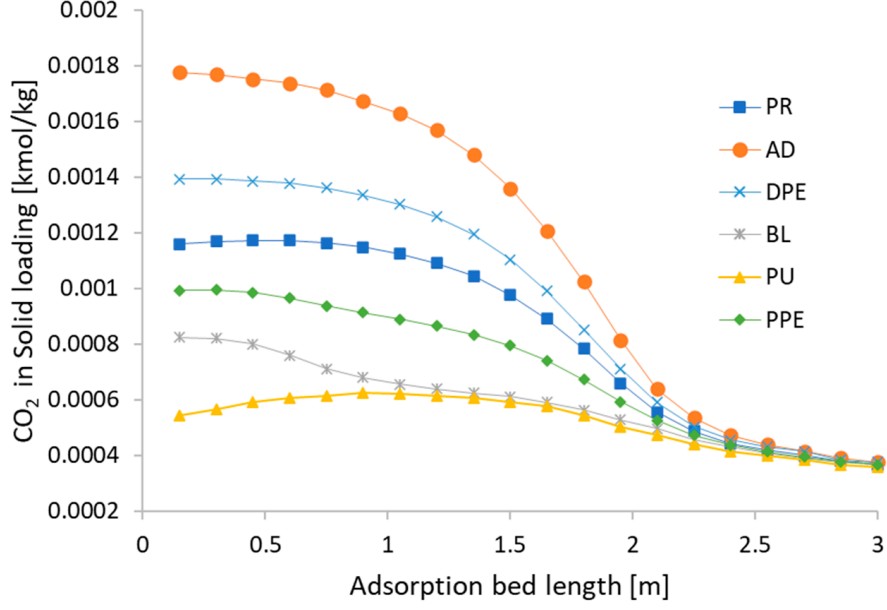

**Figure 10.** Quantity of $CO_2$ adsorbed along the column at the end of the various phases (cycle 94).

Figure 11 shows the mole fraction of $CH_4$ along the three-meter length inside the adsorbent bed in the cyclic steady state. In the PR step, $CH_4$ has a low adsorption kinetics and is not adsorbed to the adsorbent at the inlet yet. Therefore, the mole fraction equals 0.55, the same as the biogas feed. The curve shows that the mole fraction of $CH_4$ inside the adsorption bed becomes slightly higher than the biogas along the adsorption bed length. At the AD step, the mole fraction also starts at 0.55, which has the same mole fraction of biogas as $CH_4$, as the adsorbents reaches the equilibrium at the inlet, subsequently causing $CO_2$ to be no longer adsorbed. However, $CO_2$ is adsorbed when moving up to the upper part. Consequently, the $CH_4$ mole faction rises up to more than 0.92. In the DPE step, only the valve between the two beds is opened, which makes the pressure of Bed1 decrease because the product gas moves from the high pressure bed to the low pressure bed without inlet flows. Then, the mole fraction of the DPE step is shifted up and becomes higher than the mole fraction in the AD step, because the inlet valve is closed and the remaining $CO_2$ in the adsorption bed is adsorbed without input. In the BL step, the pressure in Bed1 drops to the desorption pressure and the adsorbed $CO_2$ is removed from the adsorbent. According to the Langmuir isotherm, $CO_2$ comes out when the pressure drops, which in turn affects the mole fraction of $CH_4$, leading to the lowest value among all six steps. In the PU step, due to the injection of a higher mole fraction of $CH_4$ flowing through the bed from the counter-current direction, the remaining adsorbed $CO_2$ is removed, and the mole fraction of $CH_4$ becomes higher than that in the BL step. In the PPE step, the discharged gas from another adsorption bed comes in at the top of the bed. It has a lower $CH_4$ mole fraction than the PU step, but it does not maintain its value at this step and the $CO_2$ begins re-adsorbing. Therefore, in this step, the mole fraction of $CH_4$ is almost higher than the PU step along the adsorption bed length. Figure 12 shows the results of the $CO_2$ mole fraction in direction opposition to those of $CH_4$, because we assume that the proportions of biogases include only $CH_4$ and $CO_2$.

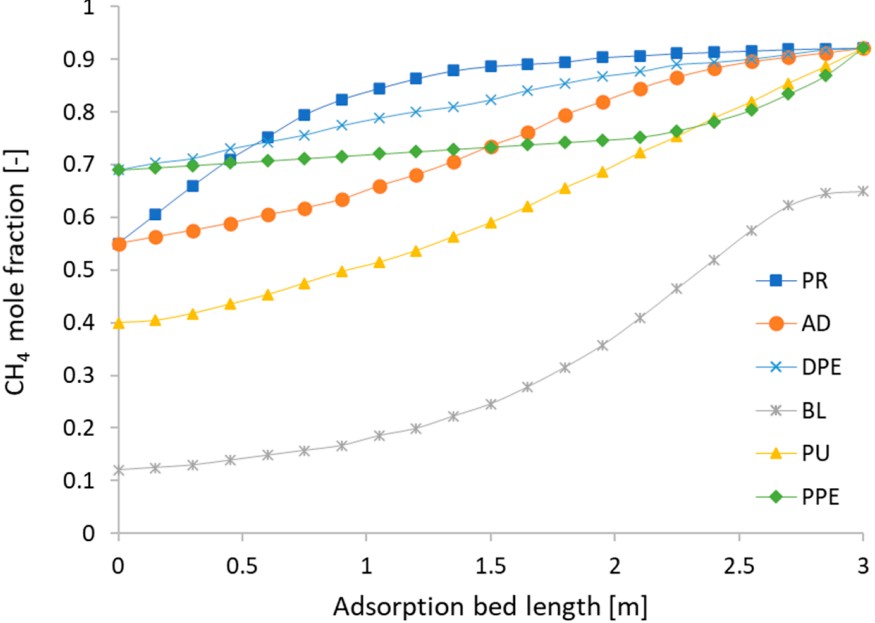

**Figure 11.** The mole fraction of $CH_4$ at the cyclic steady state with Aspen Adsorption.

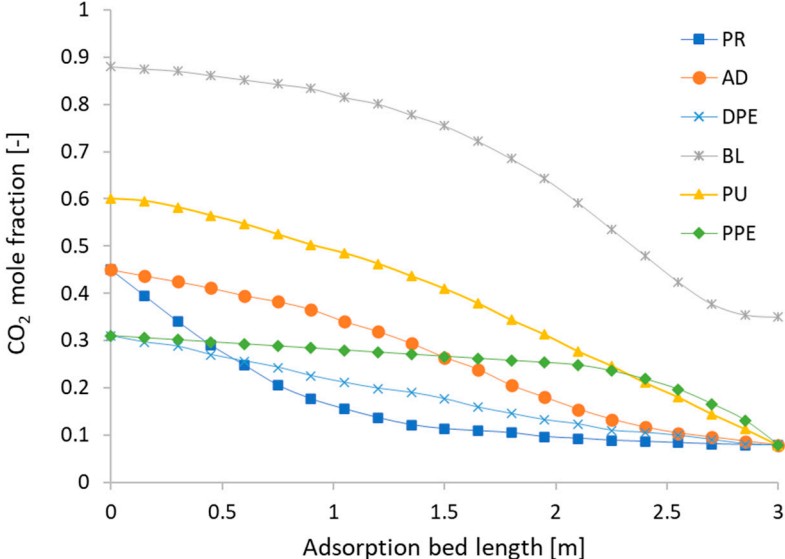

**Figure 12.** The mole fraction of $CO_2$ at the cyclic steady state with Aspen Adsorption.

This simulation makes it possible to obtain an average purity of biomethane of 92.08% over time, with a recovery of 81.63%. In addition, the results show that the real operating data fit well with the simulated results.

### 3.4. Effect of the Key Operating Conditions

### 3.4.1. Effect of Flowrate on Purity

In the real case, the adsorbent and adsorption step times of the PSA process are the most significant variables that affect the purity of biomethane, as explained above. However, it is difficult to change these variables at the commercial scale. Nevertheless, we can take into account the flowrate of biogas, which can also control the purity of biomethane. The changes in the purity according to variation of flowrate are shown in Figure 13. The results show that we can increase the purity by up to 97.92% with the same dimensions of the adsorption bed by reducing the flowrate by 60%.

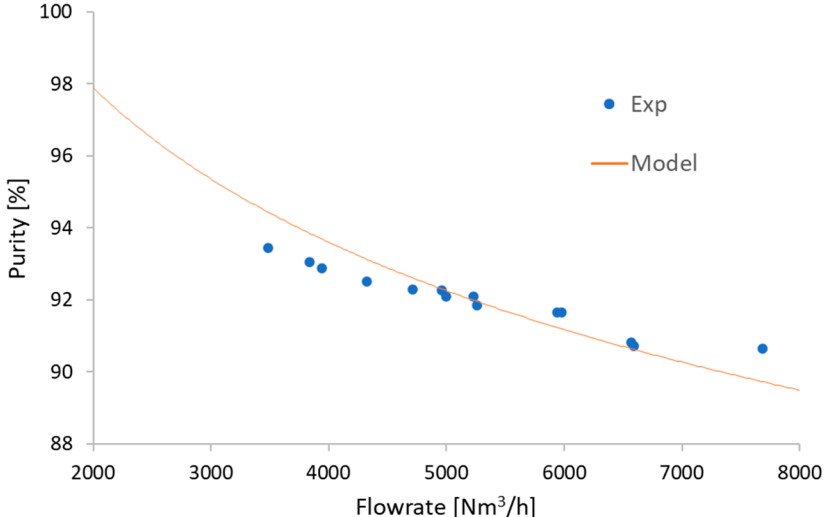

**Figure 13.** Effect of flowrate on biomethane purity.

### 3.4.2. Effect of the Adsorption Time on Purity

It has been observed in Figure 6 that when the adsorption time is longer than 300 s, the purity of $CH_4$ significantly decreases. In addition, if the adsorption time is too short, i.e., lower than 200 s, it would result in a negative effect with respect to the adsorbent limit. Therefore, this section's aim is to find the optimal adsorption time within the investigated range between 200 and 300 s. We compare the difference in purity and recovery time of methane with three adsorption step time of 200, 250 and 300 s, as shown in Table 6. It can be observed that the purity decreases and the recovery increases when increasing the adsorption time due to the interaction times of the CMS and $CO_2$. In Case 1 (200 s), the interaction between the CMS and biogas has high efficiency, as shown in Figure 6. The purity of $CH_4$ in Case 1 is higher than in Case 2 (250 s), but the difference in number of cycles in Case 1 is 23% higher than in Case 2. Consequently, the cycle time is higher and the purity is above the requirement. However, for Case 3 (300 s), the CMS was limited to absorbing only $CO_2$ and the purity was low. Therefore, the optimal condition is an adsorption step time of 250 s, which has met the minimum purity requirement and equals 92.08%. Moreover, it was very close to the real operating condition of 92.10% purity.

**Table 6.** Biomethane purity, recovery and number of cycles necessary to reach the CSS as a function of the adsorption time.

| Parameters | Case 1 | Case 2 | Case 3 |
|---|---|---|---|
| Pressure (bar) | 6.5 | 6.5 | 6.5 |
| PR (s) | 25 | 25 | 25 |
| AD (s) | 200 | 250 | 300 |
| DPE (s) | 5 | 5 | 5 |
| BL (s) | 25 | 25 | 25 |
| PU (s) | 200 | 250 | 300 |
| PPE [s] | 5 | 5 | 5 |
| Recovery of $CH_4$ (%) | 78.85 | 81.63 | 83.79 |
| Purity of $CH_4$ (%) | 92.35 | 92.08 | 91.86 |
| Number of cycles | 116 | 94 | 81 |

### 3.4.3. Effect of Pressure on Purity

With the optimal adsorption time of 250 s obtained in the previous step, we then investigate the effect of pressure inside the bed on the purity of $CH_4$. Taking into account that the real process's operating pressure is 6.5 bar, we decided to vary the pressure between 5 and 8 bar. Note that the higher the operating pressure, the larger the energy consumption. As can be seen from Table 7, an increase in pressure with the same duration of the adsorption step increases the purity but, at the same time, also decreases the recovery of $CH_4$. Therefore, the pressure was lowered as much as possible to reach a purity of 92% of the biomethane at the outlet with maximum recovery in order to reduce greenhouse gas emissions. Reducing the pressure of the input biogas can reduce the compression costs and energy consumption, however, the purity is also decreased, as illustrated in Figure 14. In addition, we have seen that the extent of purity increase is less significant when pressure increases from 6.5 to 8 bar. Therefore, the optimal pressure is estimated at 6 bar with a corresponding pressurization time (PR) of 23 s, for a total cycle time of 556 s. The speed of the incoming gas stream is 0.270 m/s, the level of purity of the biomethane obtained after 94 cycles is 92.01%, and the level of recovery is 84.11%.

**Table 7.** Biomethane purity, recovery and number of cycles necessary to reach the CSS as a function of pressure.

| Parameters | Case 1 | Optimal Case | Case 2 | Case 3 |
|:---:|:---:|:---:|:---:|:---:|
| Pressure (bar) | 5 | 6 | 6.5 | 8 |
| PR (s) | 20 | 23 | 25 | 30 |
| AD (s) | 250 | 250 | 250 | 250 |
| DPE (s) | 5 | 5 | 5 | 5 |
| BL (s) | 20 | 23 | 25 | 30 |
| PU (s) | 250 | 250 | 250 | 250 |
| PPE (s) | 5 | 5 | 5 | 5 |
| Recovery of $CH_4$ (%) | 85.32 | 84.11 | 83.45 | 81.17 |
| Purity of $CH_4$ (%) | 91.46 | 92.01 | 92.1 | 92.45 |
| Number of cycles | 98 | 95 | 94 | 92 |

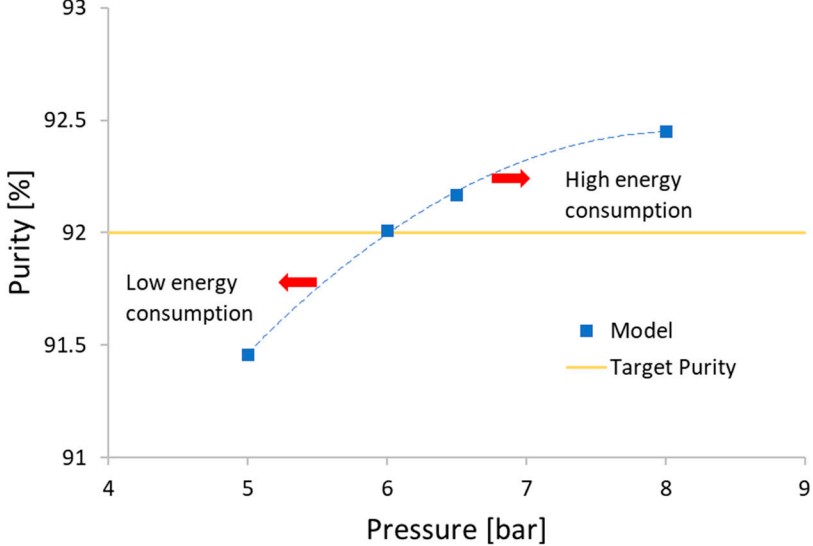

**Figure 14.** Effect of pressure on biomethane purity.

## 4. Conclusions

In this work, we simulated the pressure swing adsorption process for the separation of biogas mixed with 55% of $CH_4$ and 45% of $CO_2$. The parametric analysis was performed with inputs from a real PSA process using the Aspen Adsorption software. A number of key conclusions are drawn from this study:

- After determining the model parameter of the adsorbent, the Langmuir isotherm was used to determine the $CH_4$ and $CO_2$ isotherm of CMS-3K by using MATLAB® software with an r-square of 0.98. The mass transfer coefficients of $CH_4$ and $CO_2$ were then determined to be 0.0008 and 0.027 for the porosity of 0.33. With a set of parameters, the simulated concentration of $CO_2$ and $CH_4$ is extremely close to the experimental data measured at a temperature of 308 K. However, we found that the mass transfer coefficient changed in relation to variation in temperature.
- The kinetic parameters were regressed with MATLAB®, and by using the Arrhenius equation, the pre-exponential factor and activation energy were determined to be $1.89 \cdot 10^{27}$ s$^{-1}$ and $1.70 \cdot 10^5$ J/mol, respectively.
- The six-step PSA process was then set, in which the step time was fixed as the same as real process conditions. The adsorption and desorption pressures were 6.5 and 0.5 bar, respectively. The comparison between the simulation and real process gives a good agreement. The change

in $CH_4$ and $CO_2$ mole fractions along the adsorption bed length in a steady-state cycle followed opposite directions.

- Finally, the key operating parameters such as the flowrate, adsorption pressure and adsorption time were analyzed to find the optimal conditions. As the flowrate decreases, the purity can rise up to around 98%, while it slightly decreases when increasing the biogas flowrate. With the increase in the adsorption pressure, the amount of $CO_2$ adsorption is increased, as well as the purity of the biomethane. However, this leads to a decrease in the recovery. The purity of the biomethane increased when increasing the adsorption time until 300 s, at which point, the adsorbent performance dropped and the purity decreased. Therefore, the optimal conditions of biogas are a 5000 $Nm^3$/h flowrate, an adsorption pressure of 6 bar and an adsorption time of 250 s, which results in the purity complying to the minimum standard of 92% $CH_4$. It is worth noting that in order to obtain a comprehensive optimization, the energy consumption should be taken into account. This will be a factor of consideration in future work.

**Author Contributions:** Conceptualization, B.K. and Q.T.P.; data curation, B.K. and N.P.; funding acquisition, T.S.; investigation, B.K.; methodology, B.K.; resources, N.P.; supervision, T.S. and Q.T.P.; validation, T.S., N.P. and Q.T.P.; writing—original draft, B.K.; writing—review and editing, T.S., N.P. and Q.T.P. All authors have read and agreed to the published version of the manuscript.

**Funding:** This research was funded by Center of Excellence on Petrochemical and Materials Technology, Thailand.

**Acknowledgments:** This research has been supported by the Department of Chemical Engineering, Faculty of Engineering and the Graduate School, Kasetsart University, RE Power Service Co.,Ltd., and Center of Excellence on Petrochemical and Materials Technology.

**Conflicts of Interest:** The authors declare no conflict of interest.

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
