# Peer review of "Optimization of a Six-Step Pressure Swing Adsorption Process for Biogas Separation on a Commercial Scale"

_applsci, doi:10.3390/app10144692_

Round 1

Reviewer 1 Report

Please see attached word file.

Author Response

Thank you for your critical and useful comments/suggestions. Please see the responses in the attachment.  

Reviewer 2 Report

Dear authors,

I have now finalized reading your manuscript. It is a very interesting piece of workv and I will strongly support publication. I will make some comments about the manuscript that I hope you can consider in a revised version.

  1. Please check the language, but also the content. Lines 45-56 say almost the same as lines 65-73.
  2. Please explain what is WS and MS in Table 1.
  3. Line 46. The amount of 92% of methane is not enough according to some legislation. Although is more than enough methane to use it as vehicular fuel, to get it approved in many countries in Europe, the biogas needs to have at least 97.5% purity (or max. 2.5% CO2). The possibility of having 92% purity is if there is little inert gas (nitrogen) and some compensation with higher paraffins that will increase the Wobbe index. 
  4. Langmuir isotherms and parameters from Table 3. The parameters presented in Table 3 are given as numerical entities that are inserted in a model in Aspen adsorption. However, they were presented as part of the "Langmuir model". In such a case, IP4 is the heat of adsorption divided by the R gas constant. In such a case, adsorption is exothermic so the numbers cannot have two different signs. I know that the effect of this in the results is not very important, but that is only because you have used the assumption of "isothermal" in the operation.
  5. The option isothermal may need to be discussed. Several authors have reported temperature variations over 20K within one PSA cycle.
  6. the  increase in the diffusion coefficient can also be better described. Your model is isothermal so your curve will be a perfect and symmetric S-shape curve. The experimental curve is not: is faster (steeper) right after breakthrough and slowlier before reaching the plateau. The non-perfect S-shape comes from temperature effects: the adsorbent is heated some degrees and then diffusion gets faster (at a higher temperature). Then, the adsorbent starts to cool down and also increases its capacity, so making the curve less steep. Your model lumps the thermal effects into a mass transfer one. It is a valid and faster solution, but I think that something should be mentioned to avoid new students thinking that this is the only correct approach.
  7. Equation 8. I'm not sure if I understood correct, but you have only fitted one temperature and estimated an activation energy of adsorption. Is that correct? If that is what you have done, there is an equation with two variables fitted to one set of data which is redundant. So I'm assuming I didn't understood the procedure.
  8. General question. Did you have to tune the numerics of Aspen Adsorption or you could get convergence with the standard parameters? Can you get some info about that?
  9. Since the study deals with a "commercial" unit for biogas upgrading, I would ask the authors to please add a reference from Prof. Sircar. Although Prof. Sircar died last year, I think his family deserves the recognition of his merits. Prof. Sircar was the designer of the first PSA unit (GEMINI) for biogas upgrading. One example is here, but a publication could also make things easier to be tracked. https://patents.google.com/patent/US4770676A/en

I have no further comments / questions about the results. I think that the results obtained are very interesting and valuable and I want to congratulate the authors for the manuscript.

With my best regards,
Reviewer

Author Response

Thank you for your detailed and useful comments/suggestions. Please see the responses in the attachment.  

Reviewer 3 Report

This manuscript simulates biomethane adsorption using carbon molecular sieve. The authors used a twelve-bed, six-step PSA process which produced high purity biomethane. For the design of the adsorption bed the authors aimed at a biomethane capacity of more than 5,000 Nm3 /hr, which is close to small industrial conditions. The authors also used a Langmuir isotherm model to determine the isotherm of a mixed-gas containing 55% CH4 and 45% CO2 and experimental data to investigate the kinetic parameter and mass transfer coefficient.

This is a very well written manuscript well organised and presented that makes a significant contribution in the field and will be of interest to researchers working in the field both in industry and academia. Minor grammatical / spelling mistakes can be easily corrected. 

Author Response

Thank you for your encouraging comments. We have carefully checked and improved the language. Please check the revised manuscript.

Round 2

Reviewer 1 Report

I thank authors for the explanation and the revision of manuscript to make the readers easier to understand what the authors have done. For its current form, this article can be published.